# Promoting Physical Activity in a Primary Care Practice in People Living with Dementia and Their Family Caregivers

**DOI:** 10.3390/healthcare11091255

**Published:** 2023-04-27

**Authors:** Elena de Dios-Rodríguez, Carmen Patino-Alonso, Susana González-Sánchez, Olaya Tamayo-Morales, Joana Ripoll, Sara Mora-Simón, Jaime Unzueta-Arce, Manuel A Gómez-Marcos, Luis García-Ortiz, Emiliano Rodríguez-Sánchez

**Affiliations:** 1Unidad de Investigación en Atención Primaria de Salamanca (APISAL), Gerencia de Atención Primaria de Salamanca, Gerencia Regional de Salud de Castilla y León (SACyL), Avenida de Portugal 83, 37005 Salamanca, Spainolayatm@usal.es (O.T.-M.); jaime_ua@usal.es (J.U.-A.);; 2Instituto de Investigación Biomédica de Salamanca (IBSAL), Paseo de San Vicente, 58-182, 37007 Salamanca, Spain; 3Red de Investigación en Cronicidad, Atención Primaria y Promoción de la Salud (RICAPPS), 08007 Barcelona, Spain; 4Departamento de Estadística, Universidad de Salamanca, Calle Alfonso X el Sabio s/n, 37007 Salamanca, Spain; 5Primary Care Research Unit of Mallorca, Baleares Health Services-IbSalut, Palma, Carrer de l’Escola Graduada, 3, 07002 Palma, Spain; 6Balearic Islands Health Research Institute (IdISBa), Carrer de l’Escola Graduada, 3, 07002 Palma, Spain; 7Departamento de Psicología Básica, Psicobiología y Metodología de las Ciencias del Comportamiento, Campus Ciudad Jardín, Universidad de Salamanca, 37005 Salamanca, Spain; 8Departamento de Medicina, Universidad de Salamanca, Calle Alfonso X el Sabio s/n, 37007 Salamanca, Spain; 9Departamento de Ciencias Biomédicas y del Diagnóstico, Universidad de Salamanca, Calle Alfonso X el Sabio s/n, 37007 Salamanca, Spain

**Keywords:** caregivers, dementia, physical activity, primary care, randomized controlled trials

## Abstract

People living with dementia (PLWD) and their family caregivers report higher rates of having a sedentary lifestyle than their non-disabled peers do. This study analyzed the effectiveness of an intervention designed to increase physical activity among PLWD and their family caregivers in primary health care settings. A cluster-randomized multicenter clinical trial was conducted. Participants from four health centers were randomly assigned to the intervention group (IG) or the control group (CG) in a 1:1 ratio using Epidat software. After a seven-day period with a digital pedometer (Omron Hj-321 lay-UPS), participants were asked to complete the International Physical Activity Questionnaire Short Form (IPAQ-SF). PLWD and caregivers allocated to the IG were given brief advice, educational materials and an additional 15 min appointment to prescribe an individualized physical activity plan. Seventy PLWD and 80 caregivers were assigned to the CG and 70 PLWD and 96 caregivers were assigned to the IG. Results of the pedometer assessment show that in PLWD, the IG’s activity increased by 52.89 aerobic steps at 6 months and the CG’s activity decreased by 615.93 aerobic steps, showing a net increase in the IG of 668.82 (95% CI: −444.27 to 1781.91; *p* = 0.227). For caregivers in the IG, activity increased by 356.91 aerobic steps and in the CG it decreased by 12.95 aerobic steps, showing a net increase in favor of the IG of 369.86 (95%CI: −659.33 to 1399.05; *p* = 0.476). The effectiveness of interventions to increase physical activity in this group of people with dementia and their caregivers did not achieved positive results overall but may have provided suggestions for family physicians and physical therapists to improve physical activity among people with dementia and their families.

## 1. Introduction

Population aging has led to a rise in chronic diseases, as well as disability, resulting in greater dedication to care by family members, who in many cases become, in many cases, the main caregivers [1]. More than two-thirds of people suffering from dementia continue to live at home, significantly impacting their families [2], and worsening the quality of life of both those with dementia and their family caregivers [3,4]. These challenges are exacerbated by a shortage of dementia care specialists, which places an increasing burden on primary physicians to provide care for people living with dementia (PLWD) [5,6,7].

A caregiver provides support and care to someone else living with dementia. It is known that more than 30% of PLWD have an average of three caregiving members in their family [1]. However, most interventions focus only on the PLWD and just one caregiver [8], without considering family function [5,9]. It is known that PLWD have a less active lifestyle in comparison to their peers without health problems [10,11]. They highly depend on their caregivers to perform any physical activity [5]. As a result, caregivers have a great impact on physical activity interventions for PLWD.

Physical activity is a highly protective factor for cognitive function and a promising psychosocial strategy for the protection of cognition in older people [12,13]. However, it is difficult to measure the physical activity in this group of participants. To the best of our knowledge, practically all studies carried out on PLWD evaluate physical activity using questionnaires instead of objective measures such as the use of the pedometer [9]. Furthermore, whether or not the questionnaires are filled out by PLWD or their caregivers is not specified, so it is difficult to assess the discrepancy between performed and reported physical activity [14].

Caring for a family member living with dementia often has a negative impact on physical and psychological health [15]. Moreover, caregivers have been found to have a greater tendency towards having a sedentary lifestyle than non-caregivers do [11,16], which can exacerbate negative mental health [10] effects and cardiovascular morbidity [17,18] and increase overall mortality. It has been shown that physical activity can be effective not only as a therapy for anxiety and depression, but also as a primary prevention tool [19]. Physical activity is a beneficial intervention for healthy older people, increasing functional capacity and controlling cardiovascular disease risk factors [13,20,21].

Although physical activity interventions can have a positive impact on PLWD, they are a major challenge for caregivers. They also require caregiver involvement and can sometimes have negative side effects on PLWD, such as behavioral and psychological symptoms, pain, confusion, agitation, feeling unwell, and an increased likelihood of falls. In a review by Lamotte et al. [16] only four controlled trials that developed interventions for PLWD in dyads were considered, focusing on both PLWD and their caregivers [22,23,24,25]. The conclusion of these trials was that physical activity interventions are feasible and can have a positive effect on PLWD, promoting functional independence and facilitating caregiver care [25]. However, there is insufficient evidence of the benefits of these interventions with dyads with respect to cognitive performance and behavioral and neuropsychiatric symptoms in PLWD [17]. None of these interventions were conducted in primary care.

The Experimental Program for Physical Activity Promotion from “Programa Experimental Promoción de la Actividad Física” in Spanish (PEPAF study) [26] is the only one that has been carried out in primary care with inactive men and women of all ages to study the reduction in mortality associated with a change from being inactive to active. This study found that inactive patients who increased their physical activity, even below the minimum recommendations, significantly reduced mortality [27]. In this study, physical activity was assessed subjectively using the 7-Day Physical Activity Recall (PAR) questionnaire [28]. Incorporating methods such as those using the pedometer [29] in the measurement of physical activity in PLWD may increase the objectivity of the measurement.

Therefore, the present study aimed to evaluate the effectiveness of PEPAF intervention in primary care to increase the physical activity of PLWD and of their relative caregivers with objective and subjective measures. We also estimated the effects of the intervention in PLWD in terms of their cognitive status and level of dependence, and on caregivers in terms of mental health, overburden, and family functionality.

## 2. Materials and Methods

A field trial with two health centers (clusters) was randomly assigned to the intervention group (IG), and two more centers were assigned to the control group (CG), where normal care was to be maintained. The protocol has been published in the following sections of the methodology shown below [30] and is registered in https://clinicaltrials.gov/ct2/show/NCT02044887 (AFISDEMYF study, NCT 02044887; date: 24 January 2014). Assessments were performed at the baseline and after 6 and 12 months, between January 2016 and December 2018 (Figure 1). A pilot study was conducted for 6 months in another facility that did not participate in the final phase of the study [30].

### 2.1. Setting and Participants

This study was conducted at four primary care centers in Spain. The participants at the center were randomly assigned to the IG or CG. Health centers, rather than participants, were randomized to avoid contamination. The allocation sequence was generated in a 1:1 ratio using the Epidat software package (version 4.2; Xunta de Galicia) by an independent investigator who was blinded until the group was assigned. Due to the nature of the study, participants could not be blinded to the intervention. Based on the PLWD morbidity register of primary care physicians participating in the study, with dementia diagnosed according to the Diagnostic and Statistical Manual of Mental Disorders (DSM-IV), those who met the inclusion criteria were selected and invited to participate. Sample size was estimated to detect a difference equal to or greater than 600 steps/day (1/2 SD) between the 2 groups. Accepting a 0.05 alpha risk and a beta risk of 0.2 in a bilateral contrast, 70 participants in the control group and 70 in the intervention group were needed. The standard deviation was assumed to be 1200 steps. The dropout rate was estimated at 10%. Therefore, we considered 140 participants to be sufficient to test the hypotheses of the study.

A detailed description of the inclusion and exclusion criteria has been published in the study protocol [30]. We invited 1 to 3 family members to participate in the trial who were caregivers at least two days a week.

### 2.2. Participant Recruitment and Consent

Approval was obtained from the ethics committee of the participating centers (11 April 2013) before the start of the study, and all participants signed the informed consent form prior to inclusion in the study. Medical professionals and nurses at the participating health centers were responsible for recruiting patients and caregivers who participated in the study. When a patient visited the health center, the purpose of the study and the content of the user investigation were explained. When the patient and caregiver expressed their intention to participate, contact details were given to the investigators to arrange a schedule of interviews and observations. All data were collected with the consent of the patients and caregivers, and key insights and quotes were carefully selected from the full transcripts of the interview data.

### 2.3. Data Collection

Primary care professionals recruited the subjects and a researcher trained for the study in each participating center was in charge of collecting the anamnesis and examination data as detailed below. The data were collected in a database for subsequent analysis. Randomization was carried out by each participating health center, so it was completed prior to data collection.

### 2.4. Intervention

The intervention was carried out by health workers (physicians and nurses) who regularly cared for PLWD at the health center. Caregivers and PLWD might have different primary care physicians. In the first interview, the participating health workers assessed the morbidity, lifestyle, functionality, and care plans of the PLWD. The intervention carried out in the IG was the one recommended in the PEPAF study, consisting of an interview lasting 15–20 min, addressing possible problems regarding physical activity anticipated by the PLWD and caregiver, and negotiating a focused physical activity plan in compliance with the recommendations of the Centers for Disease Control and Prevention (CDC). The benefits of performing physical activity and international recommendations on weekly physical activity were explained. Finally, a recommendation was made for 30 min of moderate activity for five days a week or 20 min of vigorous activity for three days a week. Participants were instructed to perform physical activity autonomously, preferably by walking around their neighborhood. To support the intervention, PEPAF recommendations [26,27] were handed out in writing (in diptychs) to both the PLWD and caregivers. During the following three months, interviews lasting about 15 min were conducted every three weeks to encourage the performance of physical activity and offer support in case of any difficulties the PLWD may have encountered when exercising.

The health workers participating in the IG received a four-hour training session at the beginning of the study in the protocol for prescribing physical activity and were offered support during the study period to complete information or reinforce intervention content. No records other than those used in the primary care provided to the PLWD (medical history) were requested to avoid overloading the participants’ attention. The CG health workers provided normal care and delayed any systematic intervention in physical activity until the end of the study unless the reasons for consultation or health problems of the PLWD and caregivers were directly related to physical activity.

### 2.5. Outcome Variables

#### 2.5.1. Primary Measurement Variable

The primary outcome measure was the change in physical activity from the baseline to 6 months. The measurement of objective physical activity was carried out with a pedometer and subjective physical activity was assessed with the International Physical Activity Questionnaire Short Form (IPAQ-SF) on the same days the pedometer measurement was carried out.

(1)Digital pedometer (Omron Hj-321 lay-UPS): The pedometer was previously validated [31]. Its piezoelectric sensors use multi-position-sensing technology. It shows the total steps, aerobic steps, distance covered, and calories consumed, and stores the results of the last 7 days. The pedometer was worn by the PLWD and caregivers for 9 consecutive days in order to record measurements for 7 full days. The application was configured with the participant’s data (sex, age, weight, height, and step length).(2)The International Physical Activity Questionnaire Short Form (IPAQ-SF): The subjective physical activity record was collected for 7 days using the 9-item version of the IPAQ-SF questionnaire [32]. The IPAQ-SF is used for a general measure of physical activity and has been recognized as a valid and reliable tool. It consists of questions reflecting on the activities of the previous 7 days according to the following domains: (1) occupational physical activity; (2) transport-related physical activity; (3) housework, home maintenance and family care; (4) recreational, sport and physical leisure activities; and (5) time spent sitting. The sum of the products of the hours dedicated to each activity and the estimated energy expenditure (MET) provides an estimate of the kilocalories per kilogram used per day (kcal × kg^−1^ × d^−1^). The physical exercise dose is estimated in METs per minute per week (METS/min/week).

#### 2.5.2. Secondary Measurement Variables

The secondary outcome measures were the functional and cognitive status of the PLWD or the mental health of caregivers. Functionality was measured with the Barthel test and the Lawton and Brody test and the and cognitive status was measured with ADAS-Cog, the mini mental state examination (MMSE), and the clock drawing test. Mental health was measured with the 12-item General Health Questionnaire (GHQ-12), Family APGAR, and short-form Zarit test.

ADAS-Cog is a brief cognitive battery composed of several scales assessing memory, learning and recognition, language, visuo-constructive skills, ideational practice, and temporal–spatial orientation. Errors are counted and scoring can range from 0 (best) to 70 (worst). It is the most widely used general cognitive measure in clinical trials [33].

The mini mental state examination (MMSE) comprises 20 items and explores the functions of temporal–spatial orientation, attention, memory, language and constructive practice. The total score is a summation of all item scores, with 0 being the maximum error and 30 being the maximum success [34].

In the clock drawing test, the subject is instructed to draw a clock with all the numbers and to place the hands at ten past eleven. Visuoconstructive, visuospatial, planning and organization skills are assessed. The maximum total score is 7 points [35].

The 12-item General Health Questionnaire (GHQ-12) was used to assess perceived mental health [36]. It is a self-administered screening questionnaire designed to be used in a clinical setting to detect individuals with psychiatric disorders. The total score is obtained by adding the scores between 1 and 4 of the 12 items; the higher the score, the worse the state of mental health. The cut-off point is set at 12 points.

The Family APGAR assesses the functionality of the family through five components: adaptation, partnership, growth, affection and resolve. It consists of five questions which have five possible answers: never, rarely, sometimes, quite frequently, and almost always, these answers being scored from 0 to 4. A score of 10–12 indicates moderate dysfunction, 13–16 indicates mild dysfunction, and 17–20 indicates normal functionality [37].

The short-form Zarit test was applied to evaluate caregiver burden [38]. It consists of 7 questions with 5 possible responses (never, rarely, sometimes, quite frequently, and almost always), scored from 1 to 5, giving a total scoring range of 7–35. The cut-off point is set at 17 points, with higher scores representing overload situations.

In addition, the following sociodemographic variables were considered: age, marital status, educational level, number of people living together at home, number of living children, and caregiver occupation. Anthropometric variables (height, weight, and blood pressure) and morbidity were reported from the medical history of the health workers who regularly attended the PLWD.

The research team collected responses to different questions regarding the care received by the PLWD, the number of months that caregivers had been caring for the family member, and whether or not the caregiver and the PLWD lived in the same home.

### 2.6. Validity and Reliability

This study followed the recommendations of the CONSORT guidelines. The Epidat software (version 4.0) was used to randomize the primary care centers (two to the IG and two to the CG) to avoid possible contamination due to the interaction of participants from the same center. Randomization was performed by researchers who were not performing the assessment, and the researcher in charge of the analysis was blinded. Due to the nature of the study, participants could not be masked. Participants from both groups were able to participate freely in other activities during the intervention period and were able to continue participating in the activities they had previously started.

### 2.7. Data Analysis

Data were expressed as means and standard deviations or medians and interquartile ranges, if necessary, for continuous variables and as numbers and percentages for categorical variables. Normality was assessed using the Kolmogorov–Smirnov normality test. The chi-squared test was used to compare categorical variables. The *t*-test was used for independent measures and the Mann–Whitney U test was used to compare the baseline characteristics between the two groups of quantitative variables. In the *t*-tests performed, the homogeneity of variances was evaluated. Student’s *t*-test of paired data or the Wilcoxon test was used to analyze the changes at 6 months with respect to the baseline evaluation of the outcome variables within the same group. Comparisons of the changes in continuous variables between the intervention and control groups were performed using the two-way repeated measures ANOVA. In the hypothesis test, an α risk of 0.05 was set as the limit of statistical significance. Statistical analyses were performed using the IBM^®^ SPSS^®^ v.26 software (IBM Corp, Armonk, NY, USA).

## 3. Results

### 3.1. Baseline Characteristics of the Participants

Finally, Seventy PLWD were included in the IG, 70 were included in the CG and 176 caregivers were included in the study; 80 caregivers were included in the CG and 96 were included in the IG. Figure 1 (flow chart) shows that 48.6% of the participants initially evaluated completed the study at 6 months. The most frequent dropout cause was “did not want to continue”. The 12-month evaluation provided complete information on 22 PLWD (11 IG and 11 CG) and 35 caregivers (16 CG and 19 IG). Because of the high sample attrition (84.29% of PLWD and 80.45% of caregivers), effectiveness results were obtained with the 6-month assessment data.

One hundred and forty PLWD (median age = 82.00 (IQR: 78.00–85.00; 63.6% women)) and 176 carers (median age = 62.00 (IQR: 52.00–78.00; 72.7% women)) participated in the study. Seventy PLWD (80 caregivers) in the CG and 70 PLWD (96 caregivers) in the CG were assigned to the IG.

Table 1 and Table 2 compare the sociodemographic and clinical characteristics of the PLWD and caregivers in the IG and CG. The CG scores were worse on the Lawton–Brody and short-form Zarit tests (*p* < 0.001). No differences were observed when comparing the physical activity of the CG with that of the IG at the baseline.

### 3.2. Characteristics of Follow-Up

The second evaluation of PLWD involved 61.43% (*n* = 43) of the IG, and 35.71% (*n* = 25) of the CG (Figure 1 and Table 3), as well as 73.08% of the caregivers in the IG and 62.24% of the caregivers in the CG. A comparison of the PLWD who dropped out with those who participated in the second evaluation showed no differences in sex and physical activity, although there were differences in age: PLWD who dropped out had a higher mean age (84.07 ± 5.78 years vs. 80.42 ± 6.18; *p* = 0.016). There were no differences in sex and physical activity measures between caregivers who dropped out and those who underwent a second evaluation between the CG and the IG. However, there were differences with respect to age, with younger caregivers dropping out more in the IG (57.49 ± 14.05 vs. 64.67 ± 14.46 years; *p* = 0.020). Pedometer measurements were repeated in eight PLWD and three caregivers because they did not obtain valid records during the week (they were supposed to wear the pedometer) and six PLWD who lost the pedometer explicitly refused to put it on again.

The number of PLWD evaluations was higher when more than one caregiver was involved in the family group in both the CG (*p* = 0.007) and IG (*p* = 0.049). For 90% of the PLWD in the CG and 68.6% of PLWD in the IG, only one caregiver participated in the study. In 32.4% (*n* = 46) of the family groups, two caregivers were involved, and in 15.7% (*n* = 22), three caregivers were involved. All PLWD had 3 caregivers who participated in at least two evaluations. The highest percentage of attrition was observed in the PLWD group which had only one caregiver and belonged to the CG (62.9%; 44/70). Among the PLWD who underwent only the initial evaluation, 97.8% had a single caregiver in the CG and 85.2% had a single caregiver in the IG.

### 3.3. Changes in Physical Activity

The changes observed in the physical activity of the PLWD and caregivers were not significant (Table 4 and Table 5). The pedometer results show that in the PLWD in the IG, the number of aerobic steps increased to 52.89 at 6 months, while that for the PLWD in the CG decreased to 615.93 aerobic steps; the net increase in favor of the IG was therefore 668.82 (95% CI: −444.27 to 1781.91; *p* = 0.227).

For caregivers in the IG, the number of aerobic steps increased to 356.91 and that for those in the CG decreased to 12.95, resulting in net increases in favor of the IG of 369.86 (95%CI: −659.33 to 1399.05; *p* = 0.476) and 370.46 MVPA/min/week (IC95%: 253.34–150.15; *p* = 0.156).

### 3.4. Other Changes Measured in People Living with Dementia and Caregivers

Table 4 shows that there was an improvement in BMI for the PLWD in the IG (0.51 vs.–0.71; *p* = 0.011) but the abdominal perimeter and systolic and diastolic blood pressure in the IG were worse. Deterioration was also observed in both groups in the Barthel index (*p* < 0.05), Lawton index and in the cognitive assessment score (ADAS-cog, MMSE, clock test) but there were no differences when comparing changes between the two groups.

In caregivers (Table 5), systolic blood pressure (*p* = 0.001) and diastolic blood pressure increased in the IG. No significant changes were observed in the mental health score (GHQ-12). Improvement in the family APGAR score (*p* = 0.018) and in the short-form Zarit test score (−1.51 vs. −0.38) was observed in the IG, but the differences were not significant when comparing changes between the IG and CG.

## 4. Discussion

This study assessed the effectiveness of an intervention designed, adapted, and implemented in a primary healthcare setting to increase physical activity among PLWD and their caregivers. Looking at the changes in physical activity at six months from the baseline, the intervention was found to be ineffective. However, an analysis of the results obtained may help to develop a more effective intervention, especially in the primary care setting. For the PLWD, the pedometer assessment score showed a decrease in both the CG and IG, but results were worse in the CG. However, the activity reported using the IPAQ-SF showed poor IG results. For caregivers in both groups, there was an increase in physical activity, and better IG results were observed for both the pedometer results (total steps and aerobic steps) and vigorous activity reported on the IPAQ-SF. The effectiveness of interventions for increasing physical activity when applied in the field of primary care is inconclusive [2]; nevertheless, in the PEPAF study [20,26], improvement was achieved in those over 50 years of age. Thus, it was more effective than the results achieved in this study, although the method of measuring physical activity was different; the methods used were subjective and objective in our study and they were subjective in the PEPAF study. One plausible explanation for these differences could be that the studies were applied to different populations; the PEPAF study was applied to a healthy and sedentary population, whereas in this project, being sedentary was not an inclusion criterion. However, at the initial evaluation, only 23 caregivers (13% of the total) exceeded 10,000 steps/day, which was the recommended goal for healthy adults [39]. It was found that inactive patients, who increased their physical activity even at a minimal level, below the minimum recommendations, significantly reduced mortality [27]. Thus, the small improvement observed in our study could support the recommendation of family physicians to increase physical activity for this group of people, although it clearly needs to be improved to achieve greater efficacy.

It should also be considered that the intervention model of this project was based on the PEPAF, in which an individual intervention was performed, whereas in the AFISDEMYF study, it interventions were applied simultaneously to PLWD and their family caregivers. Thus, this cannot be classified as a typical intervention. Additionally, it cannot be classified as a group intervention because it was applied during the consultations of the health center staff with each of the family groups, consisting of one PLWD and between one and three caregivers. Most physical activity interventions in PLWD have been developed in groups in specialized dementia centers [2,8,40,41] and only four interventions have been developed with dyads (one LPLWD and one caregiver) [16]. Despite the diversity of the interventions, the 2015 Cochrane review was inconclusive in supporting these physical activity interventions in PLWD owing to the significant heterogeneity of the studies conducted [42]. The results of our study conducted in primary care are thus along the same lines, and more effective interventions need to be designed.

As more than 30% of people with dementia are cared for by three or more family caregivers [1], up to three caregivers per PLWD were invited to participate in this study. A positive feature of this approach is reflected in the fact that, when the number of caregivers is greater, they are more likely to participate in the follow up, and this circumstance is more appropriate for implementation in the primary care setting. It appears to be a promising approach to foster integrated programs that address the needs and requirements of PLWD and their caregivers in a multidimensional manner [40]. It is also probable that in people with serious illnesses such as dementia, healthcare providers pay more attention to aspects related to the morbidity of PLWD and to enhance support for caregivers, rather than implementing healthier lifestyles [2,5,43]. If this is the case, we suggest that future studies should evaluate the possibility of supporting them with physical therapists in the health center itself or with community resources in the context of interventions aimed at PLWD and their caregivers [40].

Likewise, we encountered serious difficulties in the proper use of pedometers by some PLWD, with several pedometers being lost; however, some pedometers did not record the physical activity performed on a given day. Because the participants refused to repeat the evaluations, the sample size was reduced. It was observed that discrepancies between questionnaires and objective measures were greater in people with obesity, with higher disability scores and more depressive symptoms [44]. In our study, discrepancies between the questionnaire and pedometer results were mainly found in the PLWD. This is possibly due to the fact that all questionnaire data were provided by the caregivers. In these cases, the observed discrepancies between questionnaires and objective measures may be due to different reasons than those found when comparing physical activity reported by healthy participants using questionnaires and physical activity device measurements [45]. A social desirability bias commonly causes participants to respond to physical activity verification questionnaires too optimistically, and variability in optimism towards reaching goals may influence their ability to respond accurately to questionnaires [46]. The use of accelerometers and digital devices in particular [47] can offer safer and more reliable measurements [48], and the cost of different devices must be considered, as well as considering appropriate strategies to use them correctly with this type of PLWD, so that the devices are not frequently lost. However, the possible biases related to these devices should have been the same in both the groups (the CG and IG). Therefore, the validity of the results regarding the effectiveness of the intervention was not modified. Objective measures may be the most appropriate, especially for people who are not in a position to declare reliable information [10]. With these observations, we agree with other researchers such as O.Smit et al. [45], who stated that there is still insufficient evidence for physical activity evaluation in primary care, let alone for measuring physical activity in PLWD. In practice, our results could provide suggestions for the role of objective measures to improve physical activity in PLWD.

Although it was not the aim of the study to show the physical activity levels of caregivers, in our sample, they were observed to be more sedentary than those in the study by Loi et al. [49]. Our data therefore highlight the importance of developing interventions that contribute to increasing physical activity in this population [16].

### 4.1. Other Changes Analyzed through the Intervention

In addition to evaluating the changes observed in physical activity, the impacts in several domains were evaluated. Among PLWD, no differences were observed between the IG and CG in the changes measured in the assessment of both functional and cognitive states. However, in both groups, we observed that most PLWD showed a worsening general condition. The overall prevalence of depression, anxiety and apathy is very high in all stages of the dementia population [50], so increasing physical activity in these people could also contribute to their health [19]. However, in this study we did not assess changes for a comparison between the GI and CG because there is no consensus on the method of symptom assessment and cultural differences may explain some of the variation in the prevalence of affective symptoms. However, mental health in caregivers was assessed [50].

No significant differences were found in the changes observed regarding mental health, short-form Zarit test scores, or APGAR family scores between the IG and CG caregivers. However, it should be noted that the APGAR family questionnaire score and short-form Zarit test score showed improvement for the IG, suggesting that the intervention was not harmful to the caregivers. Although no significant differences were observed, it should be noted that this improvement in the stress tests is in line with what has been observed in several interventions developed in PLWD dyads to implement physical activity [17].

### 4.2. Limitations and Strengths

The main limitation of this study was the sample’s attrition during the follow-up period, which limited the power of the statistical analysis and increased the beta risk. Although the study follow up was planned for up to 12 months, the progress of the disease itself, accompanied by a worsening of general health, meant that the number of PLWD who dropped out at 6 months was large. As this would compromise the power of the study, we decided to analyze the data for this period only. The main reasons for dropout were refusal to continue the study and failure to locate participants. These are usually the most frequent causes faced by similar population studies with older people and these are exacerbated when the degree of disability is considerable [44]. However, we have observed that participation may be improved if more than one caregiver per PLWD is included, which may represent a strength of the study’s methodology.

In the CG, there were no differences in age, sex, or physical activity level between the caregivers and PLWD who participated in both assessments, and those who dropped out of the study. However, there were differences with respect to age in the IG, in which older PLWD and younger caregivers dropped out. It is possible that caregivers under 65 years of age felt more overburdened with other activities (including work) and did not prioritize physical activity in the context of the care plan offered to the PLWD, and that older PLWD felt less motivated to implement physical activity. The fact that caregivers presented a higher level of burden was also likely to have contributed to the considerable dropout rate in the CG, since it is the only characteristic in which a difference was found between the IG and CG in the initial caregiver assessment (Table 2). To our knowledge, this is the first study to implement a primary care intervention aimed at simultaneously increasing physical activity among PLWD and their caregivers. To reach clinically relevant conclusions regarding the potentially significant effects of the intervention on physical activity and other aspects, a longer intervention period and more specific measures are required to avoid substantial dropout rates [16,44].

## 5. Conclusions

Although some positive results were found globally, no differences were found between the IG and CG in terms of the increased physical activity. In addition, no differences were observed in the functional and cognitive status of the PLWD or in the mental health of the caregivers. When the number of caregivers per PLWD was greater, they were more likely to participate in the follow up, which is appropriate for the implementation in primary care settings. In practice, our results may provide suggestions for the role of family physicians, and physical therapists in improving physical activity in PLWD and their family members.

## Figures and Tables

**Figure 1 healthcare-11-01255-f001:**
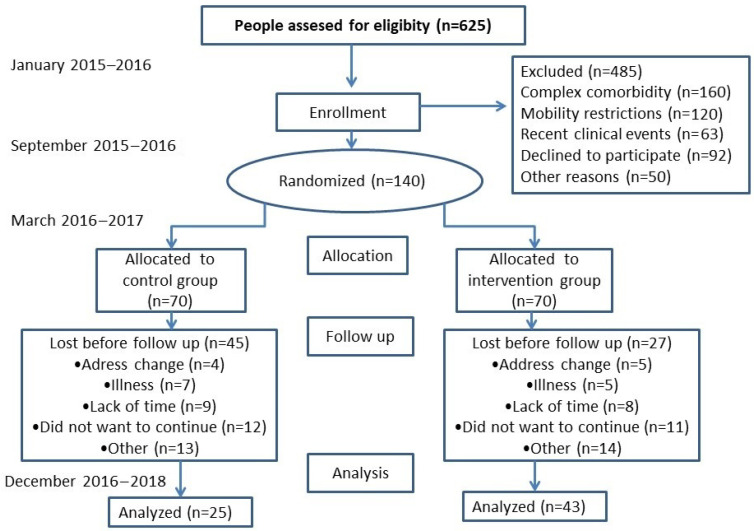
Study flowchart: enrollment of participants and completion study.

**Table 1 healthcare-11-01255-t001:** Comparison of demographic, clinical and physical activity of control and intervention groups at baseline—people living with dementia.

Variables	Control (*n* = 70)	Intervention (*n* = 70)	*p* Value
Demographic characteristics			
Age (years)	82.00 (78.00–85.00)	81.50 (78.00–86.25)	0.446
Gender: woman	41 (58.6)	48 (68.6)	0.219
Years of schooling.	8.00 (6.00–8.00)	8.00 (6.00–8.00)	0.532
Marital status: married	36 (65.5)	41 (70.7)	0.551
Lives with:			0.346
One person	53 (75.7)	48 (68.6)	
Two or more people	17 (24.3)	22 (31.4)	
Classification number of children:			0.145
No children	3 (4.3)	8 (11.4)	
One children	13 (18.6)	8 (11.4)	
Two children	17 (24.3)	24 (34.3)	
Three or more children	37 (52.2)	30 (44.8)	
Clinical characteristics:			
Abdominal perimeter (cm)	92.07 (10.96)	95.24 (14.54)	0.331
Systolic blood pressure (mmhg)	128.50 (122.50–138.50)	132.00 (118.50–145.50)	0.997
Diastolic blood pressure (mmhg)	73.00 (66.75–82.00)	74.00 (65.00–84.00)	0.652
Body mass index (kg/m^2^)	27.34 (6.28)	26.99 (4.50)	0.755
Hypertension	24 (34.3)	36 (51.4)	0.040
Hypercholesterolemia	29 (41.4)	28 (40.0)	0.863
Diabetes mellitus	12 (17.1)	13 (18.6)	0.825
Smoking	1 (3.4)	4 (6.6)	0.810
Obesity	6 (20)	16 (25.8)	0.540
Barthel index	70 (55–80)	75 (45–85)	0.639
Lawton–Brody index	1 (0–3)	3 (1–3)	0.009
ADAS-Cog	48.50 (42.00–62.75)	45.00 (39.00–60.50)	0.561
Mini mental state examination	15.44 (7.51)	18.06 (7.68)	0.095
Clock drawing test	1 (0–4)	2 (0–5)	0.483
Number of months receiving care:			0.235
Less than 18	12 (18.5)	18 (28.1)	
Between 18 and 36	13 (20.0)	18 (28.1)	
Between 37 and 68	19 (29.2)	12 (18.8)	
More than 68	21 (32.3)	16 (25.0)	
Pedometer			
Total steps/day	3340.89 (2831.53)	4384.52 (4988.75)	0.350
Aerobics steps/day	1697.97 (1695.99)	2316.95 (3011.92)	0.463
Kilocalories/day	100.01 (49.68–151.84)	111.29 (54.85–165.86)	0.273
Total Steps			0.407
Less than 7000	19 (82.6)	43 (78.2)	
Between 7000–10,000	4 (17.4)	8 (14.5)	
More than 10,000	0 (0.0)	4 (7.3)	
IPAQ-SF			
METS/min/week	1052.47 (926.56)	1412.00 (1391.72)	0.242
MVPA/min/week	16.77 (54.74)	28.88 (123.95)	0.641
Physical Activity Intensity			0.405
Light	7 (28)	22 (38.6)	
Moderate	17 (68)	30 (52.6)	
Intense	1 (4)	5 (8.8)	

Notes—IPAQ-SF: short form international physical activity questionnaire short form; MET: metabolic equivalent; MVPA: moderate–vigorous physical activity. Values expressed as mean (±standard deviation) median (IQR) or frequencies (percent). Chi-squared test and Mann–Whitney U test were used to test differences in all measures where Student’s *t*-test was applied.

**Table 2 healthcare-11-01255-t002:** Comparison of demographic, clinical and physical activity of control and intervention groups at baseline—caregivers.

Variables	Control (*n* = 78)	Intervention (*n* = 98)	*p* Value
Demographic characteristics			
Age (years)	69.00 (53.00–78.00)	59.00 (51.00–76.00)	0.120
Gender: woman	52 (66.7)	76 (77.6)	0.107
Years of schooling	9 (8–13)	12 (8–15)	0.119
Marital status: married	65 (83.3)	82 (83.7)	0.952
Lives			0.960
Alone	3 (4.1)	3 (3.3)	
With one person	40 (54.1)	48 (53.3)	
With two or more people	31 (41.9)	39 (43.3)	
Current job situation:			0.343
Homemaker	32 (41.03)	36 (36.37)	
Works	13 (16.67)	22 (24.45)	
Retired	3 (3.85)	9 (9.20)	
Does not work	30 (38.46)	31 (31.63)	
Number of children	1.98 (1.54)	1.95 (1.57)	0.648
No children	17 (21.8)	19 (19.4)	
One child	11 (14.1)	16 (16.3)	
Two children	23 (29.5)	36 (36.7)	
Three or more children	27 (34.6)	27 (27.6)	
Clinical characteristics			
Abdominal perimeter (cm)	92.24 (15.42)	90.94 (13.24)	0.567
Systolic blood pressure (mmhg)	130.00 (115.25–146.25)	120.00 (110.75–135.00)	0.026
Diastolic blood pressure (mmhg)	77.00 (70.00–86.00)	75.50 (70.00–84.00)	0.207
Body mass index (kg/m^2^)	26.76 (4.94)	26.21 (4.17)	0.430
Hypertension	26 (33.3)	32 (32.7)	0.924
Hipercholesterolemia	25 (32.1)	34 (34.7)	0.712
Diabetes mellitus	6 (7.7.)	9 (9.2.)	0.725
Smoking	14 (17.9)	18 (18.4.)	0.943
Obesity	26 (33.3)	41 (41.8)	0.248
Anxiety/depression	15 (19.2)	21 (21.4)	0.720
Mental health GHQ-12 score	3.2 (4.19)	2.26 (3.58)	0.121
Family APGAR score	13.61 (5.29)	15.11 (4.5)	0.055
Short-form Zarit test score	19.96 (6.75)	15.95 (6.41)	<0.01
Pedometer			
Total steps/day	6039.32 (3211.98)	7067.45 (3717.84)	0.067
Aerobic steps/day	1891.74 (1719.76)	2060.35 (1933.64)	0.603
Kilocaloríes/day	104.21 (47.07–164.46)	122.71 (72.43–187.43)	0.042
Total steps:			0.314
Less than 7000	45 (36.4)	46 (51.7)	
Between 7000–10,000	18 (25.4)	28 (31.5)	
More than 10,000	8 (11.30)	15 (16.9)	
IPAQ-SF:			
METS/min/week	2322.43 (1865.04)	2273.98 (2273.33)	0.883
MVPA/min/week	97.17 (190.28)	112.65 (300.41)	0.701
Physical activity intensity:			0.857
Light	11 (14.9)	12 (13.2)	
Moderate	44 (59.5)	61 (67)	
Intense	19 (25.7)	18 (19.8)	

Note—GHQ-12: general health questionnaire 12. IPAQ-SF: short form international physical activity questionnaire; MET: metabolic equivalent; MVPA: moderate–vigorous physical activity. Values expressed as mean (±standard deviation), median (IQR) or frequencies (percent). Chi-squared test and Mann–Whitney U test were used to test differences in all measures except those of the short form Zarit where Student’s *t*-test was applied.

**Table 3 healthcare-11-01255-t003:** Comparison of the number of evaluations carried out by people living with dementia with the number of participating caregivers.

Number of Participating Caregivers for Each Person Living with Dementia	Number of Evaluations Carried Out by the People Living with Dementia			
	Initial Only	6 Months	12 Months	Total
Control group: †				
1 caregiver	44 (97.8)	9 (64.3)	10 (90.9)	63 (90.0)
2 caregivers	1 (2.2)	4 (28.6)	1 (9.1)	6 (8.6)
3 caregivers	0 (0.0)	1 (7.1)	0 (0.0)	1 (1.4)
Total	45 (100)	14 (100)	11 (100)	70 (100)
Intervention group #				
1 caregiver	23 (85.2)	20 (62.5)	5 (45.5)	48 (68.6)
2 caregivers	4 (14.8)	7 (21.9)	5 (45.5)	16 (22.9)
3 caregivers	0 (0.0)	5 (15.6)	1 (9.1)	6 (8.6)
Total	27 (100)	32 (100)	11 (100)	70 (100)
Total	72 (51.4)	46 (32.9)	22 (15.7)	140 (100)

Notes: Values expressed as frequencies (percent). Chi-squared test is used. † *p* value = 0.007; # *p* value = 0.049.

**Table 4 healthcare-11-01255-t004:** Changes in people living with dementia at 6 months compared to baseline.

Variables	Control Group (*n* = 32)		Intervention Group (*n* = 35)		Mean Difference (Intervention-Control)	
	Mean (CI 95%)	*p* Value ^†^	Mean (CI 95%)	*p* Value ^†^	Mean (CI 95%)	*p* Value ^‡^
Physical Activity						
Pedometer						
Total steps (day)	−898.46 (−2225.51–428.59)	0.171	−646.37 (−1502.21–209.48)	0.134	252.10 (−1218.33–1722.53)	0.732
Aerobic steps (day)	−615.93 (−1344.33–112.47)	0.087	52.89 (−695.45–801.23)	0.883	668.82 (−444.27–1781.91)	0.227
IPAQ-SF						
MET/min/week	−148.23 (−338.46–42.00)	0.119	−258.47 (−870.87–353.93)	0.396	−110.24 (−896.31–675.84)	0.779
MVPA/min/week	−14.96 (−38.88–8.95)	0.205	−31.54 (−101.67–38.59)	0.366	−16.58 (−106.90–73.74)	0.714
Clinical characteristics:						
Abdominal perimeter (cm)	3.44 (−0.915–7.80)	0.114	−3.16 (−7.31–0.99)	0.130	−6.60 (−12.43–(−0.77))	0.031
Systolic blood pressure (mmhg)	5.95 (−3.55–15.46)	0.206	−0.73 (−8.68–7.219)	0.852	−6.69 (−19.07–5.39)	0.284
Diastolic blood pressure (mmhg)	4.50 (−1.84–10.84)	0.154	−0.70 (−5.52–4.11)	0.767	−5.20 (3.86–(−12.96))	0.184
Body mass index (kg/m^2^)	−0.71 (−1.33–0.08)	0.28	0.51 (−3.55–1.08)	0.079	1.21 (0.43–0.36)	0.006
Functional and cognitive state						
Barthel index	−15.63 (−26.18–(−5.07))	0.006	−14.64 (−25.75–(−3.54))	0.011	0.98 (−15.49–17.45)	0.906
Lawton–Brody index	−0.20 (−0.98–0.58)	0.599	0.19 (−0.59–0.96)	0.625	0.39 (−0.75–1.52)	0.496
ADAS-Cog	5.84 (−3.25–14.93)	0.194	3.00 (−5.51–11.51)	0.477	−2.84 (−15.54–9.85)	0.655
MMSE	−1.29 (−2.64–0.07)	0.061	−1.80 (−2.97–(−0.63))	0.004	−0.51(−2.31–1.28)	0.569
Clock drawing test	0.05 (−0.90–1.01)	0.909	−0.20 (−1.02–0.62)	0.620	−0.25 (−1.50–0.99)	0.685

Notes—IPAQ-SF: short form international physical activity questionnaire; MET: metabolic equivalent. MVPA: moderate–vigorous physical activity. MMSE: mini mental state examination. Student’s *t*-test for apparent data was used to measure intragroup differences and two-way repeated measures ANOVA was used to evaluate the difference between groups. *p* value: ^†^ differences between physical activity at 6 months compared to baseline; ^‡^ differences between changes intervention group and control group.

**Table 5 healthcare-11-01255-t005:** Changes in caregivers at 6 months compared to baseline.

Variables	Control Group (*n* = 55)		Intervention Group (*n* = 60)		Mean Difference (Intervention-Control)	
	Mean (CI 95%)	*p* Value ^†^	Mean (CI 95%)	*p* Value ^†^	Mean (CI 95%)	*p* Value ^‡^
Physical Activity						
Pedometer						
Total steps (day)	377.23 (−218.28–972.73)	0.208	569.41 (−565.57–568.38)	0.319	192.18 (−1109.25–1493.60)	0.770
Aerobic steps (day)	−12.95 (−566.78–540.89)	0.962	356.91 (−378.17–407.39)	0.351	369.86 (−659.33–1399.05)	0.476
IPAQ-SF						
MET/min/week	609.55 (−53.36–1272.46)	0.071	545.25(−420.38–1510.89)	0.262	−64.29 (−592.34–1240.09)	0.914
MVPA/min/week	124.68 (−153.75–403.11)	0.358	495.14 (−59.33–1049.62)	0.074	370.46 (253.45–150.51)	0.156
Clinical characteristics						
Abdominal perimeter (cm)	4.04 (0.59–7.49)	0.023	2.55 (−0.26–5.37)	0.075	−1.48(2.212–(−5.90))	0.506
Systolic blood pressure (mmhg)	−3.71 (−8.85–1.42)	0.152	6.18 (2.55–9.82)	0.001	9.90 (3.09–3.77)	0.002
Diastolic blood pressure (mmhg)	0.13 (−3.29–3.56)	0.939	9.09 (−0.47–4.27)	0.114	1.76(2.04–(−2.28))	0.390
Body mass index (kg/m^2^)	1.51 (−0.3–3.39)	0.111	0.23 (−0.31–0.78)	0.391	−1.28 (0.93–<8–3.13)	0.172
Mental Health						
Mental healthGHQ-12 score	0.40 (−0.82–1.63)	0.512	0.74 (−0.45–1.94)	0.218	0.34 (−1.36–2.04)	0.691
Family APGAR score	0.02 (−1.12–1.16)	0.970	1.38 (0.24–2.52)	0.018	1.36 (−0.23–2.95)	0.093
Short-form Zarit test score	−0.38 (−1.77–1.00)	0.582	−1.51 (−3.44–0.42)	0.123	−1.13 (−3.48–1.22)	0.344

Notes: IPAQ-SF International Physical Activity Questionnaire -Short Form; MET: metabolic equivalent. MVPA: moderate-vigorous physical activity; MMSE: mini mental state examination. Student’s *t*-test for apparent data was used to measure intragroup differences and two-way repeated measures ANOVA was used to evaluate the difference between groups. *p* value: ^†^ differences between physical activity at 6 months compared to baseline; ^‡^ differences between changes intervention group and control group.

## Data Availability

The data that support the findings of this study are openly available at http://hdl.handle.net/10366/151082 (accessed on 1 January 2023) or can be obtained by contacting the authors.

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
