# Peer review of "Promoting Physical Activity in a Primary Care Practice in People Living with Dementia and Their Family Caregivers"

_healthcare, 2023, doi:10.3390/healthcare11091255_

Round 1
Reviewer 1 Report
Dear esteemed authors,
This study presents a highly compelling investigation into the application of physical activity for Persons Living With Dementia (PLWD) and their caregivers in a primary healthcare setting. Despite the limitations imposed by a substantial dropout rate, these challenges reflect the practical realities inherent in primary healthcare environments.
I strongly recommend that future research considers integrating digital devices, such as smartwatches, to measure physical activity levels in PLWD and their caregivers within primary healthcare settings. Such devices can potentially provide more precise and accurate data, leading to more meaningful results in understanding the relationship between physical activity and health outcomes for PLWD and their caregivers.
Consequently, I recommend that forthcoming studies implement digital devices to measure physical activity and evaluate their potential benefits in this context. This approach may result in more effective interventions and strategies aimed at enhancing the health and well-being of PLWD and their caregivers.
I am pleased to extend my congratulations to the authors and express my strong endorsement of the publication of this article in its present form. The methodology employed in this study is well-considered and meticulously explained, while the analysis is presented with remarkable clarity and precision. Furthermore, the conclusion appropriately reflects the study's findings. The article is written in an accessible and eloquent style, free of any linguistic or grammatical errors, and exhibits no indications of plagiarism or any form of improper citation. As such, it is my great pleasure to offer my recommendation for the acceptance of this article.
Author Response
We would like to thank you for reviewing the manuscript.
The authors are very grateful for your comments. We hope that the publication of the article may motivate the development of interventions to increase physical activity in this group of people.
We would also like to thank you for the recommendation to use smartwatches and inform you that we are already using it in some studies such as the Evident age. (BMC Geriatrics 2019; 19:19)
Reviewer 2 Report
Dear Authors,
Thank you for your submission.
I enjoyed reviewing your manuscript. Overall, it is well-written and covers an important topic.
Please make the following minor edits:
Abstract:
please add p values for in-between groups differences
please add what mHealth refers to.
line 168: provide more information about the questionnaire that was created for the purpose of this study (i.e., who developed the survey, how was it validated, data collection, etc.)
I look forward to reviewing your revised manuscript.
Many thanks

Author Response
Author's Reply to the Review Report (Reviewer 2)
Manuscript ID: ijerph-2237730
Type of manuscript: Article
Title: Promoting physical activity in a primary care practice in people living with dementia and their family caregivers
Please provide a point-by-point response to the reviewer’s comments and either enter it in the box below or upload it as a Word/PDF file. Please write down "Please see the attachment." in the box if you only upload an attachment.
Comments and Suggestions for Authors
Dear Authors,
Thank you for your submission.
I enjoyed reviewing your manuscript. Overall, it is well-written and covers an important topic.
Comments and Suggestions for Authors
Dear Authors,
Thank you for your submission.
I enjoyed reviewing your manuscript. Overall, it is well-written and covers an important topic.
Authors' Answer
We would like to thank you for reviewing the manuscript.
The authors are very grateful for your comments.
Please make the following minor edits:
1.-Abstract: please add p values for in-between groups differences
Authors' Answer
We have modified the wording of the paragraph, to facilitate the compression of the data shown, remaining as follows:
“Results of pedometer assessment show that in PLWD, the IG increased 52.89 steps/aerobic at 6 months and CG decreased 615.93 steps/aerobic, a net increase by the IG of 668.82 (95% CI: -444.27 to 1781. 91; p=0.227). In caregivers IG increased 356.91 steps/aerobic and CG decreased 12.95 steps/aerobic, a net increase in favor of IG of 369.86 (95%CI: -659.33 to 1399.05; p=0.476).”
2.-please add what mHealth refers to.
Authors' Answer
We have removed the term mHealth, because strictly speaking, the intervention performed does not meet the criteria to be considered in that category.
3.-line 168: provide more information about the questionnaire that was created for the purpose of this study (i.e., who developed the survey, how was it validated, data collection, etc.)
Authors' Answer
We have added the following information on the Methods in the Variables and measurement instruments section:
Variables and measurement instruments
2.4. Outcome Variables
2.4.1. Primary Measurement Variable
The primary outcome measure was the change in physical activity from baseline to 6 months. The measurement of objective physical activity was carried out with a pedometer and subjective physical activity was assessed with the international physical activity questionnaire short form (IPAQ-SF) covering the same days as the pedometer.
1) Digital pedometer (Omron Hj-321 lay-UPS): The pedometer was previously validated [31]. Its piezoelectric sensors use multi-position sensing technology. It shows total steps, aerobic steps, distance covered and calories consumed, and stores the results of the last 7 days. The pedometer was worn by PLWD and caregivers for 9 consecutive days in order to record 7 full days. The application was configured with the participant's data (sex, age, weight and height, step length).
2) The international physical activity questionnaire short form (IPAQ-SF): The subjective physical activity record was collected for 7 days using the 9-item version of the IPAQ-SF questionnaire [32]. The IPAQ-SF is a general measure of physical activity which has been recognized as a valid and reliable tool. It consists of questions reflecting on the activities of the previous 7 days according to domain: 1) occupational physical activity; 2) transport-related physical activity; 3) housework, home maintenance and family care; 4) recreational, sport and physical leisure activities; and 5) time spent sitting. The sum of the products of the hours dedicated to each activity and the estimated energy expenditure (MET) provides an estimate of the kilocalories per kilogram used per day (kcal * kg-1 * d-1). The physical exercise dose is estimated in METs per minute per week (METS/min/week).
2.4.2. Secondary Measurement Variables
Secondary outcome measures were the functional and cognitive status of LPWD or in the mental health of caregivers. Functionalitywas measured with the Barthel test and the Lawton and Brody test and the and cognitive status with ADAS-Cog, Mini Mental State Examination (MMSE), Clock Drawing Test. The mental health measured with 12-item General Health Questionnaire (GHQ-12), Family APGAR, and Short-form Zarit Test.
ADAS-Cog is a brief cognitive battery composed of several scales assessing memory, learning and recognition, language, visuo-constructive skills, ideational practice, and temporal-spatial orientation. Errors are counted and scoring can range from 0 (best) to 70 (worst). It is the most widely used general cognitive measure in clinical trials [33].
Mini–Mental State Examination (MMSE) comprises 20 items and explores the functions of temporal-spatial orientation, attention, memory, language and constructive practice. The total score is a summation of all item scores, with 0 being the maximum error and 30 being the maximum success [34].
In the clock drawing test, the subject is instructed to draw a clock with all the numbers and to place the hands at ten past eleven. Visuo-constructive, visuospatial, planning and organization skills are assessed. The maximum total score is 7 points [35] .
The 12-item General Health Questionnaire (GHQ-12) was used to assess perceived mental health [36.]. It is a self-administered screening questionnaire designed to be used in a clinical setting to detect individuals with psychiatric disorders. The total score is obtained by adding the scores between 1 and 4 of the 12 items: the higher the score, the worse the state of mental health. The cut-off point is set at 12 points.
The Family APGAR assesses the functionality of the family through five components: Adaptation, Partnership, Growth, Affection and Resolve. Its of five questions have five possible answers: never, rarely, sometimes, quite frequently, almost always, scored from 0 to 4. A score of 10-12 indicates moderate dysfunction, 13-16 mild dysfunction, and 17-20 normal functionality [37].
The short-form Zarit test was applied to evaluate the caregiver burden [38]. It consists of 7 questions with 5 possible responses (never, rarely, sometimes, quite frequently, almost always), scored from 1 to 5, giving a total scoring range of 7-35. The cut-off point is set at 17 points, with higher scores representing overload situations.
In addition, the following sociodemographic variables were considered: age, marital status, educational level, number of people living together at home, number of living children, and caregiver’s occupation. Anthropometric variables (height, weight, and blood pressure)and morbidity were reported from the medical history of health workers who regularly attended PLWD.
The research team collected responses to different questions regarding the care received by the PLWD, the number of months that caregivers had been caring for the family member, and whether the caregiver and the PLWD lived in the same home.
4.- I look forward to reviewing your revised manuscript.
Many thanks.

Reviewer 3 Report
Response to Authors
Thanks to the authors and reviewers for allowing me to read this manuscript. The study is interesting and potentially novel, given available research to date. However, it has some major shortcomings that must be corrected before the paper can be published. I encourage the researchers to patiently revise their manuscript and bring it to the best possible standard. The specific comments I have for the authors are as follows:
Abstract
1. There are serious grammar and language errors in this section. For example, the phrase “A single” before “People Living with Dementia” should probably not be in the section. Moreover, the entire section is not coherent and is difficult to read. The authors should ask for professional language editing of their manuscript.
2. Though the authors previously published the study protocol, it is necessary for details about the design to be provided in the abstract and in the main work. For example, information about randomisation and selection should be provided as done in randomised controlled trials.
3. The authors mentioned results on mortality, but I didn’t see this variable being measured in the manuscript. Were you trying to write morbidity? Please revise this and be consistent with what you report.
4. What do the numbers in the parentheses in the abstract represent?
Introduction
Please use the abbreviation PA consistently after it was first used in the manuscript. You alternated between PA and physical activity, which is not a good way to write.
What did you mean by the word “overload”?
Your introduction should tell us why you measured PA both objectively and subjectively. Why not use only the objective method? If this was done to improve the evidence, then justify it.
Your last sentence under the introduction seems to be reporting the results; it is too early to report the results. Please delete.
Methods and Materials
Please remove “The Materials”
As mentioned, give us some more information about the design, though the protocol has been published.
Why did you allocate patients and caregivers in two facilities to the control group and those in other facilities to the experimental group? Please justify this. I thought this biased the randomisation process. You could have put all together and randomly assigned the participants to the two groups in a 1:1 ratio.
The phrase “sample losses” should be “attrition”; this is the scientific word to use.
The standard attribution rate is 20%, so why did you use 10%? As you can see, the actual attrition rate was more than 10%.
I did not see any measure on mortality.
How did you handle missing data?
What did you use the chi-square test for; I don’t seem to see its relevance in the analysis.
“X2 test” used should be replaced with “chi-square test” if you want to keep this tool in the analysis.
You said normality was assessed, but you did not state whether it was achieved. Please clearly indicate.
Another assumption that governs the t-test is the homogeneity of variances assumption. This was not mentioned and assessed.
If the data were normally distributed, the best statistical tool would be the two-way repeated measures ANOVA; this tool gives you the opportunity to concurrently examine differences in PA between the two groups as well as at baseline and at 6 months follow-up. The current analysis is possibly biased because of the groups and timelines being separated in the analysis. If the data were not normally distributed, then the non-parametric version of two-way repeated measures ANOVA should be used. Therefore, I don’t think you need a chi-square test.
Please some limitations have not been acknowledged. For example, the average PA scores of the two groups appear to be different, but these differences were not significant possibly because of the small sample size due to high attrition. I believe the differences would have been significant with a larger sample. Thus, the high attrition rate reduced the statistical power of the tests. Please discuss this as a limitation.
Conclusions
You measured several secondary outcomes, but I didn’t see some of them in the analysis and conclusion.
Author Response
Reviewer 3
Manuscript ID: healthcare-2237730
Type of manuscript: Article
Title: Promoting physical activity in a primary care practice in people living with dementia and their family caregivers.
Comments and Suggestions for Authors
Response to Authors
Thanks to the authors and reviewers for allowing me to read this manuscript. The study is interesting and potentially novel, given available research to date. However, it has some major shortcomings that must be corrected before the paper can be published. I encourage the researchers to patiently revise their manuscript and bring it to the best possible standard.
Authors' Answer
We would like to thank you for reviewing the manuscript.
The authors are very grateful for your comments.
The specific comments I have for the authors are as follows:
Abstract
1.There are serious grammar and language errors in this section. For example, the phrase “A single” before “People Living with Dementia” should probably not be in the section. Moreover, the entire section is not coherent and is difficult to read. The authors should ask for professional language editing of their manuscript.
Authors' Answer
Sentence has been deleted “A single” before “People Living with Dementia”.
Following the indications of reviewer, we have edited in English language the manuscript. We attach an edition certificate.
- Though the authors previously published the study protocol, it is necessary for details about the design to be provided in the abstract and in the main work. For example, information about randomisation and selection should be provided as done in randomised controlled trials.
Authors' Answer
We have added some details about the methodology (randomization and intervention) in the abstract:
“A cluster-randomized multicenter clinical trial was conducted. Participants from four health centers were randomly assigned to the intervention group (IG) or the control group (CG) in a 1:1 ratio using Epidat software. After a seven-day period with a digital pedometer (Omron Hj-321 lay-UPS), participants were asked to complete the International Physical Activity Questionnaire Short Form (IPAQ-SF). PLWD and caregivers allocated to the IG were given brief advice and educational materials and an additional 15-minute appointment to prescribe an individualized physical activity plan. Seventy PLWD and 80 caregivers were assigned to the CG and 70 PLWD and 96 caregivers to the IG.”.
- The authors mentioned results on mortality, but I didn’t see this variable being measured in the manuscript. Were you trying to write morbidity? Please revise this and be consistent with what you report.
Authors' Answer
We have deleted in the abstract the effects on mortality that were achieved with the PEPAF intervention, since the objectives of this study were not to assess mortality.
- What do the numbers in the parentheses in the abstract represent?
Authors' Answer
4.a.We have modified the wording referring to the participants (the number of caregivers was indicated in parentheses), deleting the parentheses to better understand the distribution of the intervention and control groups:
“Seventy PLWD and 80 caregivers were assigned to the CG and 70 PLWD and 96 caregivers to the IG.”
4.b. In the results paragraph, we have modified the wording of the paragraph to facilitate the understanding of the data shown, as follows:
“ Results of pedometer assessment show that in PLWD, the IG increased 52.89 steps/aerobic at 6 months and CG decreased 615.93 steps/aerobic, a net increase by the IG of 668.82 (95% CI: -444.27 to 1781. 91; p=0.227). In caregivers IG increased 356.91 steps/aerobic and CG decreased 12.95 steps/aerobic, a net increase in favor of IG of 369.86 (95%CI: -659.33 to 1399.05; p=0.476).”
5.- Introduction
Please use the abbreviation PA consistently after it was first used in the manuscript. You alternated between PA and physical activity, which is not a good way to write.
Authors' Answer
The entire manuscript has been revised to change the abbreviation to physical activity.
6.-What did you mean by the word “overload”?
Authors' Answer
Sorry, it is a mistake in translation. We changed to caregiver overburden.
7.-Your introduction should tell us why you measured PA both objectively and subjectively. Why not use only the objective method? If this was done to improve the evidence, then justify it.
Authors' Answer
The intervention model was the PEPAF study (2003-2007), in which it was shown that the prescription of physical exercise by the family doctor increased physical activity as assessed using the 7-Day Physical Activity Recall (PAR) questionnaire. Therefore, the idea in this study was to maintain this same strategy. However, when this project was designed, the importance of making objective measurements using mobile devices was already beginning to be highlighted, but it was not yet well established which device was the most appropriate for showing the changes made by the elderly in relation to physical activity, since they can take walks, which is easy to measure with pedometers. Therefore, we decided to incorporate this method of measuring physical activity as well.
We have added the following paragraph in the introduction (line 102):
The Experimental Program for Physical Activity Promotion from the ”Programa Experimental Promoción de la Actividad Física,” in Spanish (PEPAF study) [26] is the only one that has been carried out in primary care with inactive men and women of all ages to study the reduction in mortality associated with a change from inactive to active. This study found that inactive patients who increased their physical activity, even below the minimum recommendations, significantly reduced mortality [27]. In this study, physical activity was assessed subjectively using the 7-Day Physical Activity Recall (PAR) questionnaire [28] Incorporating methods such as the pedometer [29] in the measurement of physical activity in PLWD may increase the objectivity of the measurement.
- Sallis JF, Haskell WL, Wood PD, et al. Physical activity assessment methodology in the Five-City Project. Am J Epidemiol. 1985;121(1):91-106
- Cisek-Woźniak A, Mruczyk K, Wójciak RW. The Association between Physical Activity and Selected Parameters of Psychological Status and Dementia in Older Women. Int J Environ Res Public Health. 2021 Jul 15;18(14):7549.
8.-Your last sentence under the introduction seems to be reporting the results; it is too early to report the results. Please delete.
Authors' Answer
We agree with the reviewer's suggestion, so we have deleted the last sentence of the introduction.
9.-Methods and Materials
Please remove “The Materials”
Authors' Answer
“The materials” have been deleted.
- As mentioned, give us some more information about the design, though the protocol has been published.
Authors' Answer
We have added in the main work the following information on the Methods section:
- In the first section:
“A pilot study was conducted for six months in another facility that did not participate in the final phase of the study [30].”
2.4. Outcome Variables
2.4.1. Primary Measurement Variable
The primary outcome measure was the change in physical activity from baseline to 6 months. The measurement of objective physical activity was carried out with a pedometer and subjective physical activity was assessed with the international physical activity questionnaire short form (IPAQ-SF) covering the same days as the pedometer.
1) Digital pedometer (Omron Hj-321 lay-UPS): The pedometer was previously validated [31]. Its piezoelectric sensors use multi-position sensing technology. It shows total steps, aerobic steps, distance covered and calories consumed, and stores the results of the last 7 days. The pedometer was worn by PLWD and caregivers for 9 consecutive days in order to record 7 full days. The application was configured with the participant's data (sex, age, weight and height, step length).
2) The international physical activity questionnaire short form (IPAQ-SF): The subjective physical activity record was collected for 7 days using the 9-item version of the IPAQ-SF questionnaire [32]. The IPAQ-SF is a general measure of physical activity which has been recognized as a valid and reliable tool. It consists of questions reflecting on the activities of the previous 7 days according to domain: 1) occupational physical activity; 2) transport-related physical activity; 3) housework, home maintenance and family care; 4) recreational, sport and physical leisure activities; and 5) time spent sitting. The sum of the products of the hours dedicated to each activity and the estimated energy expenditure (MET) provides an estimate of the kilocalories per kilogram used per day (kcal * kg-1 * d-1). The physical exercise dose is estimated in METs per minute per week (METS/min/week).
2.4.2. Secondary Measurement Variables
Secondary outcome measures were the functional and cognitive status of LPWD or in the mental health of caregivers. Functionalitywas measured with the Barthel test and the Lawton and Brody test and the and cognitive status with ADAS-Cog, Mini Mental State Examination (MMSE), Clock Drawing Test. The mental health measured with 12-item General Health Questionnaire (GHQ-12), Family APGAR, and Short-form Zarit Test.
ADAS-Cog is a brief cognitive battery composed of several scales assessing memory, learning and recognition, language, visuo-constructive skills, ideational practice, and temporal-spatial orientation. Errors are counted and scoring can range from 0 (best) to 70 (worst). It is the most widely used general cognitive measure in clinical trials [33].
Mini–Mental State Examination (MMSE) comprises 20 items and explores the functions of temporal-spatial orientation, attention, memory, language and constructive practice. The total score is a summation of all item scores, with 0 being the maximum error and 30 being the maximum success [34].
In the clock drawing test, the subject is instructed to draw a clock with all the numbers and to place the hands at ten past eleven. Visuo-constructive, visuospatial, planning and organization skills are assessed. The maximum total score is 7 points [35] .
The 12-item General Health Questionnaire (GHQ-12) was used to assess perceived mental health [36.]. It is a self-administered screening questionnaire designed to be used in a clinical setting to detect individuals with psychiatric disorders. The total score is obtained by adding the scores between 1 and 4 of the 12 items: the higher the score, the worse the state of mental health. The cut-off point is set at 12 points.
The Family APGAR assesses the functionality of the family through five components: Adaptation, Partnership, Growth, Affection and Resolve. Its of five questions have five possible answers: never, rarely, sometimes, quite frequently, almost always, scored from 0 to 4. A score of 10-12 indicates moderate dysfunction, 13-16 mild dysfunction, and 17-20 normal functionality [37].
The short-form Zarit test was applied to evaluate the caregiver burden [38]. It consists of 7 questions with 5 possible responses (never, rarely, sometimes, quite frequently, almost always), scored from 1 to 5, giving a total scoring range of 7-35. The cut-off point is set at 17 points, with higher scores representing overload situations.
In addition, the following sociodemographic variables were considered: age, marital status, educational level, number of people living together at home, number of living children, and caregiver’s occupation. Anthropometric variables (height, weight, and blood pressure) and morbidity were reported from the medical history of health workers who regularly attended PLWD.
The research team collected responses to different questions regarding the care received by the PLWD, the number of months that caregivers had been caring for the family member, and whether the caregiver and the PLWD lived in the same home.
- Why did you allocate patients and caregivers in two facilities to the control group and those in other facilities to the experimental group? Please justify this. I thought this biased the randomisation process. You could have put all together and randomly assigned the participants to the two groups in a 1:1 ratio.
Authors' Answer
In the study design we decided that all PLWD from each primary care center would be in the same group (intervention or control) to avoid possible contamination between participants from the same center. Only physicians from the IG centers were trained on the study protocol, counselling and physical activity prescription.
In the new version of the manuscript, we have modified the section on randomization to better understand the process as follows:
- “In 2.1. Setting and Participants:
“This study was conducted at four primary care centers in Spain. The participants at the center were randomly assigned to IG or CG. Health centers, rather than participants, were randomized to avoid contamination The allocation sequence was generated in a 1:1 ratio using the Epidat software package (version 4.2; Xunta de Galicia) by an independent investigator and was blinded until the group was assigned. Due to the nature of the study, participants could not be blinded to the intervention. Based on the PLWD morbidity register of primary care physicians participating in the study, with dementia diagnosed according to the Diagnostic and Statistical Manual of Mental Disorders (DSM-IV), those who met the inclusion criteria were selected and invited to participate. Seventy PLWD were included in the IG and 70 in the CG. A detailed description of the inclusion and exclusion criteria has been published in the study protocol [30]. We invited 1 to 3 family members to participate in the trial who were caregivers at least two days a week. Finally, 176 caregivers were included in the study: 80 in the CG and 96 in the IG. Figure 1 (flow chart) shows that 48.6% of the participants initially evaluated completed the study at 6 months.
The most frequent dropout cause was "did not want to continue". The 12-month evaluation provided complete information on 22 PLWD (11 IG and 11 CG) and 35 caregivers (16 CG and 19 IG). Because of high sample attrition (84.29% of PLWD and 80.45% of caregivers), effectiveness results were obtained with the 6 months assessment data.”
- The phrase “sample losses” should be “attrition”; this is the scientific word to use.
Authors' Answer
We have changed sample “losses" to "attrition”.
- The standard attrition rate is 20%, so why did you use 10%? As you can see, the actual attrition rate was more than 10%.
Authors' Answer
Although the standard attrition rate is 20%, we do not have a reference directly related to interventions carried out in primary care centers with PLWP and their relatives. A pilot study was previously conducted in another primary care center with only 30 PLWD and we are confident that the number of attrition would be lower. When the study was designed, we expected to achieve a attrition rate of 10%, since these were very dependent on the health system, but reality has shown that the attrition have been much higher than estimated. At the end of the study, we have accepted that the number of people who gave up on completing the study was excessive. For this reason, we have emphasized that in future studies it is important to find ways of facilitating participation in these interventions we propose that more than one family member per PLWD be invited to participate.
14.I did not see any measure on mortality.
Authors' Answer
The aim of this study was not to measure mortality, so this variable was not used. The sentence in the abstract referring to it has been deleted.
15.How did you handle missing data?
Authors' Answer
Missing data were considered missing, no data imputation was performed.
16.What did you use the chi-square test for; I don’t seem to see its relevance in the analysis. “X2 test” used should be replaced with “chi-square test” if you want to keep this tool in the analysis.
Authors' Answer
The "chi-square test" was used in Tables 1, 2 and 3, to compare qualitative variables between different groups, so it is named in statistical methods.
- You said normality was assessed, but you did not state whether it was achieved. Please clearly indicate.
Authors' Answer
In Tables 1 and 2, quantitative variables that did not have a normal distribution were expressed as median and inter-quartile range.
- Another assumption that governs the t-test is the homogeneity of variances assumption. This was not mentioned and assessed.
Authors' Answer
In the t-tests performed, the homogeneity of variances was evaluated, using the statistical significance as it corresponds to the result of this test.
- If the data were normally distributed, the best statistical tool would be the two-way repeated measures ANOVA; this tool gives you the opportunity to concurrently examine differences in PA between the two groups as well as at baseline and at 6 months follow-up. The current analysis is possibly biased because of the groups and timelines being separated in the analysis. If the data were not normally distributed, then the non-parametric version of two-way repeated measures ANOVA should be used. Therefore, I don’t think you need a chi-square test.
Authors' Answer
We repeated the analysis using the two-way repeated measures ANOVA to compare the differences between the baseline and 6-month measurements between the two groups, as only the effect at 6 months was analyzed. Therefore, being a comparison of two measures, baseline and 6 months, the results obtained are the same as with the t-Student for independent data, between the differences of 6 months and baseline, which is how the comparison was made. In any case, we added the two-way repeated measures ANOVA in the statistical analysis and in the foot of the tables, since we agree with the reviewer that it is a more appropriate way of performing the analysis. The presentation of the data separately in columns 1 and 3 of tables 4 and 5 are merely informative of the behavior of each group. Column 5 compares the changes obtained in each group between them, to obtain the magnitude and confidence interval. As mentioned above, we have changed the significance analysis and performed it with two-way repeated measures ANOVA, as indicated by the reviewer. The chi-square test was only used in the baseline comparison (Tables 1, 2 and 3) of qualitative variables.
Therefore, we have modified the statistical analysis section to read as follows:
“Data are expressed as means and standard deviations or medians and interquartile ranges, if necessary, for continuous variables and as numbers and percentages for categorical variables. Normality was assessed using the Kolmogorov–Smirnov normality test. The chi-square test was used to compare categorical variables. T-test for independent measures and Mann-Whitney U test were used to compare baseline characteristics between the two groups of quantitative variables. In the t-tests performed, the homogeneity of variances was evaluated. Student's t-test of paired data or Wilcoxon test was used to analyze the changes at 6 months with respect to the baseline evaluation of the outcome variables within the same group. Comparisons of the changes of continuous variables between groups of intervention and control were performed using the two-way repeated measures ANOVA. In the hypothesis test, an α risk of 0.05 was set as the limit of statistical significance. Statistical analyses were performed using the IBM® SPSS® v.26 software (IBM Corp, Armonk, NY, USA).”
- Please some limitations have not been acknowledged. For example, the average PA scores of the two groups appear to be different, but these differences were not significant possibly because of the small sample size due to high attrition. I believe the differences would have been significant with a larger sample. Thus, the high attrition rate reduced the statistical power of the tests. Please discuss this as a limitation.
Authors' Answer
We have added among the limitations the high number of attritions which decreases the power of the study, as follows:
“The main limitation of this study was the sample attrition during the follow-up period, which limited the power of the statistical analysis and increased the beta risk”.
- Conclusions
You measured several secondary outcomes, but I didn’t see some of them in the analysis and conclusion.
Authors' Answer
The conclusion has been modified to read as follows:
“Although some positive results were found globally, no differences were found between the IG and CG in terms of the increased physical activity. Nor were differences observed in the functional and cognitive status of PLWD or in the mental health of caregivers. When the number of caregivers per PLWD was greater, they were more likely to participate in the follow-up, which is appropriate for implementation in primary care settings. In practice, our results may provide suggestions for the role of family physicians, and physical therapists in improving physical activity in PLWD and their family members. “
